# Aldosterone-Mediated Sodium Retention Is Reflected by the Serum Sodium to Urinary Sodium to (Serum Potassium)^2^ to Urinary Potassium (SUSPPUP) Index

**DOI:** 10.3390/diagnostics10080545

**Published:** 2020-07-30

**Authors:** Evelien Kanaan, Matthias Haase, Oliver Vonend, Martin Reincke, Matthias Schott, Holger S. Willenberg

**Affiliations:** 1Division for Specific Endocrinology, University Hospital Dusseldorf, Medical Faculty HHU Dusseldorf, D-40225 Dusseldorf, Germany; Evelien.Kanaan@uk-essen.de (E.K.); Matthias.Haase@med.uni-duesseldorf.de (M.H.); Matthias.Schott@med.uni-duesseldorf.de (M.S.); 2Department of Nephrology, University Hospital Dusseldorf, Medical Faculty HHU Dusseldorf, D-40225 Dusseldorf, Germany; vonend@nephrologie-wiesbaden.de; 3Medizinische Klinik und Poliklinik IV, Klinikum der Universität, Ludwig-Maximilians-Universität München, D-80336 Munich, Germany; martin.reincke@med.uni-muenchen.de; 4Division of Endocrinology and Metabolism, Rostock University Medical Center, D-18057 Rostock, Germany

**Keywords:** adrenal, kidney, hypertension, aldosterone, renin, hypokalemia

## Abstract

The serum sodium to urinary sodium ratio divided by the (serum potassium)^2^ to urinary potassium ratio (SUSPPUP formula) reflects aldosterone action. We here prospectively investigated into the usefulness of the SUSPPUP ratio as a diagnostic tool in primary hyperaldosteronism. Parallel measurements of serum and urinary sodium and potassium concentrations (given in mmol/L) in the fasting state were done in 225 patients. Of them, 69 were diagnosed with primary aldosteronism (PA), 102 with essential hypertension (EH), 26 with adrenal insufficiency (AI) and 28 did not suffer from the above-mentioned disorders and were assigned to the reference group (REF). The result of the SUSPPUP formula was highest in the PA group (7.4, 4.2–12.3 L/mmol), followed by EH (3.2, 2.3–4.3 L/mmol), PA after surgery (3.9, 3.0–6.0 L/mmol), REF (3.4 ± 1.4 L/mmol) and AI (2.9 +/− 1.2 L/mmol). The best sensitivity in distinguishing PA from EH was reached by multiplication of the aldosterone to renin-ratio (ARR) with the SUSPPUP formula (92.7% at a cut off > 110 L/mmol), highest specificity was reached by the SUSPPUP determinations (87.2%). The integration of the SUSPPUP ratio into the ARR helps to improve the diagnosis of hyperaldosteronism substantially.

## 1. Introduction

Sodium retention is regulated by aldosterone which is secreted in dependency of potassium, angiotensin II and other stimuli [1]. Therefore, disturbed potassium and sodium chloride sensing in *zona glomerulosa* cells may result in autonomous secretion of aldosterone [2,3,4,5,6,7,8], termed primary aldosteronism (PA), which leads to sodium retention and through active mechanisms of water conservation to hypervolemic hypertension [9]. The conservation and dilution of sodium happens on the expense of potassium and protons which led to the classical description of PA as hypokalemic hypertension with metabolic alkalosis [10]. The inverse relation of sodium and potassium concentrations in the serum and the urine of individuals affected by PA is reflected by the sodium to urinary sodium ratio divided by the (serum potassium)^2^ to urinary potassium (SUSPPUP) ratio [11]. In a follow-up study, it was shown to aid in the diagnosis of PA, especially if this value is combined with the aldosterone to renin ratio, ARR [12]. However, a large retrospective study suggested that serum potassium measurements have a higher specificity than SUSPPUP calculations [13]. This study was conducted to evaluate the SUSPPUP formula in different clinical settings in a prospective way.

## 2. Patients and Methods

The Else Kröner-Fresenius German Conn’s registry (Ethical committee # 206-07 for Munich and 3027 for Dusseldorf, approved on 18 December 2007 and 2 April 2008) and the steroid hormones in patients with pituitary, adrenal or gonadal endocrinopathies (SHIP-PAGE) study enabled us to analyze data of patients who underwent the investigation for a disorder in sodium and/or potassium salt homeostasis (Ethical committee # A2016-0088). Blood and 24-h urine samples were collected from patients who gave their informed consent between December 2007 and December 2013. The study cohort included patients who underwent screening for secondary causes of hypertension, patients with diagnosed adrenal insufficiency and normotensive patients without the above-mentioned diseases who were assigned to a »reference group.

The diagnosis of PA was based on the guidelines of the Endocrine Society [14] using self-determined cut-off values for the sodium infusion and the fludrocortisone-suppression tests, as described previously [15]. To establish the diagnosis EH, other secondary causes of hypertension, including PA, pheochromocytoma/paraganglioma and renal artery stenosis (via contrast magnetic resonance imaging or duplex ultrasound) were excluded. Cushing’s syndrome or relevant co-secretion of cortisol was ruled out if a dexamethasone test rendered a normal result of cortisol <50 nmol/L (1.8 µg/dL).

The screening parameters collected for every patient were serum aldosterone (DPC Siemens, Bad Nauheim, Germany), the serum active renin concentration (DiaSorin, Saluggia, Italy), the SUSPPUP ratio, calculated from measurements of sodium and potassium in serum and urine at the same time by applying the following formula: serum sodium/urinary sodium divided by (serum potassium)^2^/urinary potassium. In addition, blood pressure at the time of sampling was documented as well as the patient’s antihypertensive medication. Blood sampling was performed in following manner: It was drawn in the morning, patients were seated for at least 30 min, they had been allowed to get up and move before that 30-min period. Hemolysis was tried to be avoided by careful and restricted use of the tourniquet and rapid blood sample procession.

Screening of patients was performed without changing antihypertensive medication when such a procedure seemed not feasible or was not done by patients. Adjustment of medication prior to diagnostic confirmation was performed according to the guidelines [15]. If blood pressure required antihypertensive medication, preferably the calcium channel blocker verapamil and the alpha-adrenoceptor antagonist doxazosin were used.

The urinary electrolyte values were measured in urine collected over 24 h in most cases. If not possible, it was determined in spontaneous urine, first miction and fasting state. For a subset of patients (*n* = 39), repeated measurements were available. Bland-Altmann blots were calculated to show variability of laboratory studies. Patients had been asked to stop supplementation with potassium during urinary sampling.

The diagnostic utility was investigated by creating receiver operating characteristic curves for SUSPPUP, Potassium, ARR and ARR×SUSPPUP. Calculations of the area under the curve (AUC), the sensibility, specificity and positive and negative predictive values for optimal cut-off values were performed. For this, PA patients formed the case group, EH patients formed the control group, as the essential hypertensives are the most common patients to be distinguished from patients with PA.

Of the patients with an aldosterone-producing adenoma, 27 have been adrenalectomized and data pairs were available at baseline and after surgery as well after an anti-mineralocorticoid treatment with spironolactone, eplerenone, triamterene or amilorid—the latter two also in combination with hydrochlorothiazide. Since WHO’s daily defined doses are not suitable to calculate equivalent doses in patients with PA, the procedure is to start with 25 mg of spironolactone, followed by an increase to 50 mg in the absence of side effects or a combination with amiloride mono whenever possible or in combination with a thiazide if a switch to eplerenone is not possible. In many such cases, family doctors then frequently change to triamterene/thiazide—starting at 25/12.5 mg which we increased to 50/25 mg if renin remains low or hypokalemia persists. To investigate into the influence of other medication, PA patients and EH patients were assigned to different groups who were compared to one another.

Data were expressed as means (±standard deviation) and analyzed with a two-tailed *t*-test if the Kolmogorov–Smirnov normality test was passed. When the test for a Gaussian distribution was rejected, data were expressed as medians (interquartile range) and analyzed with a Mann–Whitney-test for independent samples and a Wilcoxon test for paired samples. 

## 3. Results

The Else Kröner-Fresenius German Conn’s registry (Ethical committee #206-07 for Munich and #3027 for Dusseldorf, approved on 18 December 2007) and the steroid hormones in patients with pituitary, adrenal or gonadal endocrinopathies (SHIP-PAGE) study enabled us to analyze data of patients who underwent the investigation for a disorder in sodium and/or potassium salt homeostasis (Ethical committee #A2016-0088, approved on 1 June 2016).

All in all, 225 consecutive patients were included into the study: 69 of them were diagnosed with PA (of them 38 with an unilateral form), 102 had essential hypertension (EH), 26 were treated for adrenal insufficiency (AI) and 28 normotensive patients were assigned to the reference group (REF). Basic characteristics of the study populations are shown in Table 1. Both blood pressure values were significantly lower in AI and REF patients than in the other two patient groups, while systolic, diastolic blood pressure and age did not significantly differ between PA and EH subjects. However, patients in the reference group were significantly younger than in the other groups (Table 1).

Hypokalemia was noted in 56.5% of PA patients and 4.9% of EH patients when the lower normal limit was assumed to be 3.7 mmol/L. However, receiver operating curve analysis showed best distinction between patients with PA and EH was made at <3.9 mmol/L (Figure 1). Defining this value as a cut-off for serum potassium, hypokalemia was present in 71.0% of patients with PA, 11.8% of patients with EH and in 1 patient of the REF group (3.6%). Patients with unilateral PA were significantly more likely to experience hypokalemia than PA patients with bilateral disease (81.6% vs. 58.1%, *p* < 0.05). However, the performance of assays showed considerable variations that were smallest for serum potassium (<25%) and greatest for the ARR and the ARR×SUSPPUP (>150%) determinations (Figure 2).

As shown in Table 2, serum potassium was significantly lower in PA patients as compared to the other diagnostic groups. However, the urinary potassium concentrations differed not much although they were highest in comparison to serum potassium concentrations as expressed by the SUSPPUP ratios in PA individuals (*p* < 0.0001). Interestingly, urinary sodium concentrations were lowest in patients with PA which was significant. Results of electrolyte determinations, hormonal measurements and statistical analyses are given in Table 2 and Table 3.

The combination of ARR and SUSPPUP showed the best performance to detect PA in a mixed cohort of patients with EH and PA and was followed by the ARR and potassium (Table 4). ARR SUSPPUP was elevated in 94.7% of individuals with unilateral and in 90.3% with bilateral PA (*p* = n.s.), the ARR detected 92.1% unilateral and 80.6 bilateral PA patients (*p* = n.s.), and the SUSPPUP index 71.1% patients with unilateral vs. 67.7% with bilateral disease (*p* = n.s.). Over the cohort of all 225 consecutive patients, the serum potassium and the SUSPPUP ratio correlated moderately positively with the ARR (Spearman-*r* = 0.484 and 0.473, *p* < 0.001).

Treatment with anti-mineralocorticoids and/or potassium-sparing diuretics and adrenalectomy was associated with a significant increase in serum potassium by a significant reduction in relative potassium diuresis as expressed by the SUSPPUP formula and a normalization of the ARR (Table 5, Figure 3). Blood pressure values were lower after intervention and correlated well with baseline measurements (Table 5). Blood pressure indices were lower after treatment despite a reduction in medication (2.4 at baseline vs. 2.4 on anti-mineralocorticoid treatment vs. 1.7 after surgery, *p* < 0.01).

No differences in serum potassium, ARR, SUSPPUP and ARR×SUSPPUP indices were found between PA patients who took betablockers or not, angiotensin-converting enzyme inhibitors or not, angiotensin receptor blockers or not or PA patients who were on diuretics or not (Table 6). In patients with EH, serum potassium was significantly and accompanied by an increase in renin and aldosterone lower when treated with diuretics. Significant differences were visible in EH patients when treated with angiotensin-converting enzyme inhibitors or angiotensin receptor blockers (renin-driven), that interfered mainly with the ARR (Table 6).

## 4. Discussion

This is the first prospective evaluation of the SUSPPUP ratio and we found it to correlate with the ARR whereby the correlation of serum potassium performs as well if not better. Potassium was shown to outperform the application of SUSPPUP calculations in a retrospective analysis [13]. A reason for this may be the fact that aldosterone does not only reclaim renal sodium on the expense of potassium but also leads to hypervolemia that prevents serum sodium concentrations to rise and dilutes serum potassium which is less densely concentrated in the extracellular space [9]. Therefore, serum potassium performed best if the cut-off to detect primary aldosteronism was set at <3.9 mmol/L which is above routine usage and guideline recommendations [16,17]. This may also be the cause to assume that the majority of patients affected by PA are normokalemic [18,19,20]. However, we could show that serum potassium along with SUSPPUP and renin normalizes during pharmacologic treatment and—even more and along with the ARR—after surgery, indicating that salt preservation on the expense of potassium is reversed with anti-mineralocorticoid treatment or removal of aldosterone-producing adenomas. Here, measurements of SUSPPUP and potassium were superior to the ARR in showing that anti-mineralocorticoid agents ameliorate sodium retention and hypervolemia by aldosterone. Whether SUSPPUP ratio or potassium maintain their informative power in patients with PA and kidney injury remains open but seems likely in patients with impairment in renal electrolyte handling.

The best parameter to diagnose PA was the multiplication of SUSPPUP and the ARR and confirms previous retrospectively obtained results in a different cohort of patients [12]. The same combination was also the best test to document the reversal of PA in adrenalectomized patients. Thus, the advantage of this index is to reflect an aldosterone concentration which is elevated relative to renin along with an expression of hormone action. Interestingly, the SUSPPUP ratio correlated with the response to spironolactone in patients with resistant hypertension [21].

Next to the broad landscape of influential factors, including dietary salt and potassium consumption as well as different assays in divergent study cohorts [9], we saw a remarkable intraindividual variation which was greater for hormonal test results than for the determination of electrolyte-derived indices, including SUPPUP. Interestingly, the ARR was more prone to interference with medication than the SUSPPUP index or serum potassium although this problem was visible in patients with EH only. Thus, the influence of antihypertensive medication in patients with PA was, as indicated in previous studies, rather small (discussed in more in detail elsewhere [9]).

Of note, the response to treatment was also visible by the drop in systolic and diastolic blood pressure although it did not normalize fully and required treatment in the majority of patients. Interestingly, blood pressure values at baseline correlated well with blood pressure values after therapy what may point to influential factors other than aldosterone only or may include salt consumption habits [22,23].

As a limitation, the dataset is manageable if it comes to the number of patients with essential hypertension which is rather small in comparison to their share in the general population or primary care settings which is due to the study design.

## 5. Conclusions

The combination of the SUSPPUP ratio as a marker for renal sodium retention and hypervolemia on the expense of potassium with the ARR helps to improve the diagnosis of hyperaldosteronism. The determination of the SUSPPUP index is a cheap and easy improvement in the screening and follow-up of PA patients. In addition, identification and quantification of renal potassium loss may serve the clinician also in the differential diagnosis of hypokalemia for other reasons.

## Figures and Tables

**Figure 1 diagnostics-10-00545-f001:**
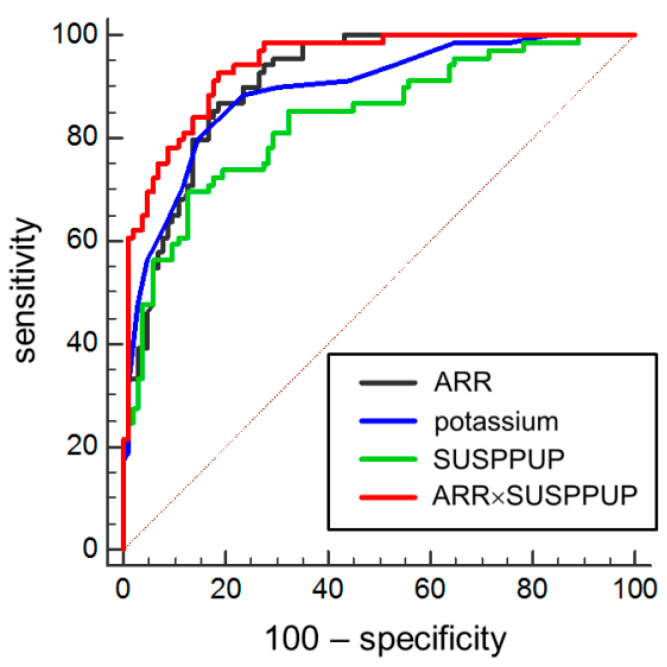
Receiver operating curve analysis for sensitivity and specificity of thresholds for serum potassium (K_S_, blue), the serum sodium to urinary sodium divided by (serum potassium)^2^ to urinary potassium (SUSPPUP) ratio (green), the aldosterone to renin ratio (ARR, black) and the product of both (red) to diagnose primary aldosteronism. Red dot line is line of equality or random chance.

**Figure 2 diagnostics-10-00545-f002:**
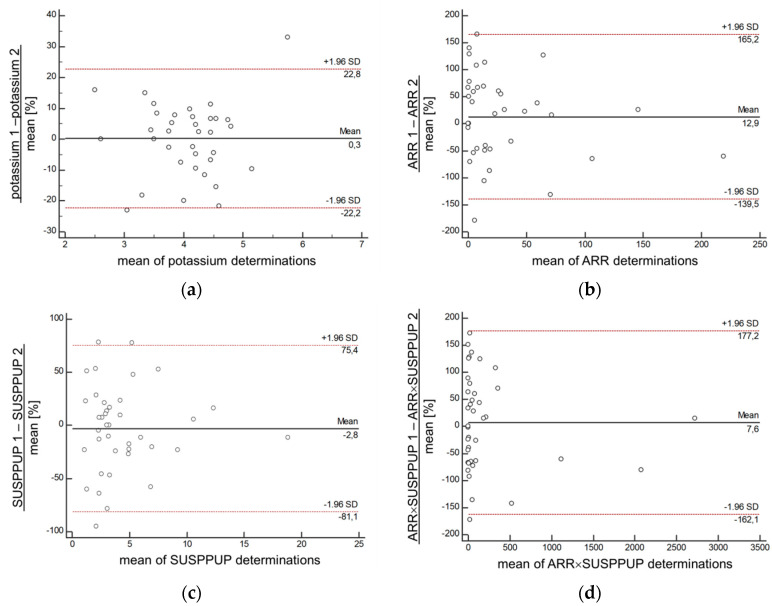
This figure shows Bland–Altman-plots to illustrate the difference between two determinations of serum potassium (**a**), the aldosterone to renin ratio (**b**), the serum sodium to urinary sodium divided by (serum potassium)^2^ to urinary potassium (SUSPPUP) ratio (**c**) and the product of both (**d**). Red dot line is line of 1.96 standard deviations.

**Figure 3 diagnostics-10-00545-f003:**
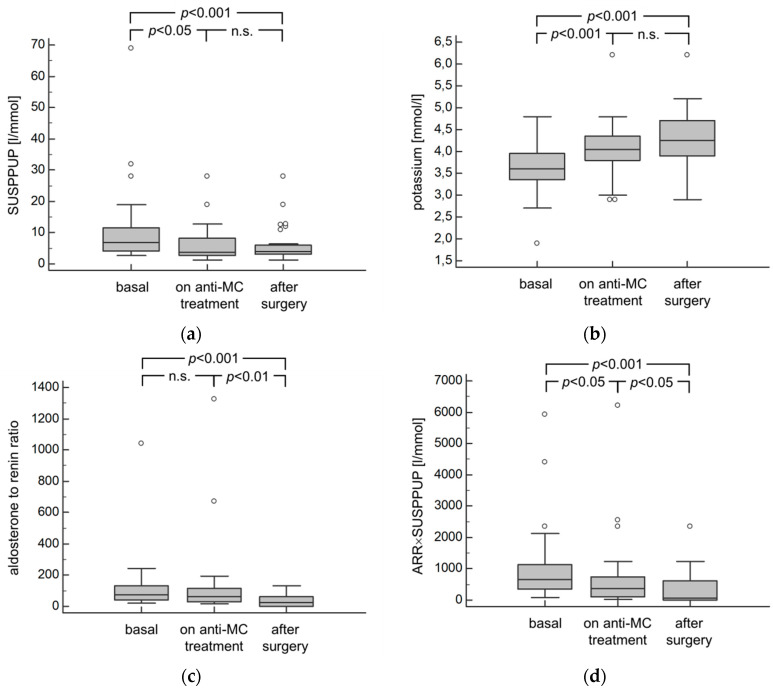
Panels show the change in the SUSPPUP ratio (**a**), serum potassium (**b**), the ARR (**c**) and ARR×SUSPPUP (**d**) from baseline, over anti-mineralocorticoid (MC) treatment to a visit between 1 and 3 months after adrenalectomy in patients with PA. Plasma renin concentrations were 5.7 ± 5.7 pg/mL on anti-MC treatment (please, see also the Patients and Methods section).

**Table 1 diagnostics-10-00545-t001:** Main patient characteristics.

Parameter	PA	EH	AI	REF
number, n	69	102	26	28
mean age [years]	53.8	53.9	51.5	40.9
females, n	35 (50.7%)^2^	69 (67.6%)	20 (76.9%)	21 (75.0%)
SBP [mmHg]	155.9 ± 20.7	150.1 ± 19.1	125.5 ± 22.4	131.9 ± 19.3
DBP [mmHg]	89.8 ± 12.5	89.0 ± 10.2	76.6 ± 13.4	79.9 ± 11.4

Abbreviations: AI—patients with adrenal insufficiency, EH—patient’s essential hypertension, PA—patients with primary aldosteronism, REF—patients assigned to the reference group, SBP—systolic blood pressure, DBP—diastolic blood pressure. Baseline characteristics given as patient numbers (percentage) or mean ± standard deviation.

**Table 2 diagnostics-10-00545-t002:** Laboratory results.

Parameter	PA	EH	REF	AI
Na_S_ [mmol/L]	143.0 (142.0–144.2)	142.0 (140.0–143.0)	140.0 (140.0–142.0)	142 (139.0–143.0)
Na_U_ [mmol/L]	88.0 ± 51.2	128.5 (83.0–168.0)	145.1 ± 68.4	125 (83.0–158.0)
K_S_ [mmol/L]	3.6 (3.3–3.9)	4.3 ± 0.4	4.3 (4.2–4.4)	4.3 ± 0.4
K_U_ [mmol/L]	56.1 ± 25.6	48.0 (37.0–73.0)	55.0 (44.0–75.0)	50.7 ± 24.3
SUSPPUP [L/mmol]	7.4 (4.2–12.3)	3.2 (2.3–4.3)	3.4 ± 1.4	2.9 ± 1.2
aldosterone [ng/L]	161.0 (108.5–252.7)	64.0 (40.0–106.0)	57.0 (33.0–94.0)	11.0 (11.0–28.0)
renin [ng/L]	1.9 (1.2–4.2)	8.9 (2.8–26.0)	5.8 (3.6–12.5)	114 ± 391.0
ARR [ng/L: ng/L]	71.7 (40.2–132.7)	9.1 (2.1–23.6)	7.5 (4.1–20.0)	1.9 (0.3–4.1)
ARR×SUSPPUP [L/mmol]	495.5 (232.9–1131.8)	28.8 (5.8–84.5)	23.3 (12.3–46.0)	3.4 (1.3–11.5)

This table shows the results of electrolyte determinations, hormonal analyses and derived indices in patients with primary aldosteronism (PA), essential hypertension (EH), adrenal insufficiency (AI) and patients assigned to the reference group (REF). Results are presented as means ± standard deviation if normally distributed or medians (interquartile range). Abbreviations: ARR—aldosterone to renin ratio, K_S_—serum potassium, K_U_—urinary potassium, Na_S_—serum sodium, Na_U_—urinary sodium, SUSPPUP—serum sodium to urinary sodium divided by (serum potassium)^2^ to urinary potassium ratio.

**Table 3 diagnostics-10-00545-t003:** Results of the statistical analyses.

Parameter	PA vs. EH	PA vs. REF	PA vs. AI	EH vs. REF	EH vs. AI	REF vs. AI
K_S_	*p* < 0.0001	*p* < 0.0001	*p* < 0.0001	n.s.	n.s.	n.s.
Na_S_	*p* < 0.0001	*p* < 0.0001	*p* < 0.0001	n.s.	n.s.	n.s.
Na_U_	*p* < 0.0001	*p* < 0.0001	*p* < 0.0001	n.s.	n.s.	n.s.
aldosterone	*p* < 0.0001	*p* < 0.0001	*p* < 0.0001	n.s.	*p* < 0.0001	*p* < 0.0001
renin	*p* < 0.0001	*p* < 0.0001	*p* < 0.0001	n.s.	n.s.	n.s.
SUSPPUP	*p* < 0.0001	*p* < 0.0001	*p* < 0.0001	n.s.	n.s.	n.s.
ARR	*p* < 0.0001	*p* < 0.0001	*p* < 0.0001	n.s.	*p* < 0.0001	*p* < 0.0001
ARR×SUSPPUP	*p* < 0.0001	*p* < 0.0001	*p* < 0.0001	n.s.	*p* < 0.0001	*p* < 0.0001

This table shows the results of the statistical comparisons of serum potassium (K_S_), serum sodium (Na_S_), aldosterone, renin, the serum sodium to urinary sodium divided by (serum potassium)^2^ to urinary potassium (SUSPPUP) ratio, the aldosterone to renin ratio (ARR) and the product of both in patients with primary aldosteronism (PA), essential hypertension (EH), adrenal insufficiency (AI) and patients assigned to the reference group (REF).

**Table 4 diagnostics-10-00545-t004:** Cut-off values, sensitivities and specificities as suggested by ROC curve analysis.

Parameter	AUC	Optimal Cut-Off	Sensitivity [%]	Specificity [%]	PPV [%]	NPV [%]
ARR	0.910	28.2	87.0	81.4	75.9	90.2
K_S_	0.891	3.9	79.7	85.3	78.3	85.3
SUSPPUP	0.835	5.25	69.6	87.2	78.7	80.9
ARR×SUSPPUP	0.942	110.3	92.7	81.4	77.1	94.3

This table shows the thresholds derived by receiver-operating curve analysis for serum potassium (K_S_), the aldosterone to renin ratio (ARR), the serum sodium to urinary sodium divided by (serum potassium)^2^ to urinary potassium (SUSPPUP) ratio and the product of both in a mixed cohort of patients with primary aldosteronism and essential hypertension.

**Table 5 diagnostics-10-00545-t005:** Pairwise analysis of parameters of baseline and post-interventional studies.

Parameter	Systolic BP [mmHg]	Diastolic BP [mmHg]	Serum Potassium [mmol/L]	ARR	SUSPPUP [L/mmol]	ARR×SUSPPUP [L/mmol]
baseline	154.9 ± 43.5	91.1 ± 13.4	3.6 ± 0.6	74 (42.2–136.5)	7.2 (4.9–12.2)	718 (380–1121)
after surgery	143.3 ± 19.1	87.0 ± 12.7	4.2 ± 0.7	23 (2.0–63.0)	3.9 (3.0–6.0)	46 (6–567)
correlation	*r* = 0.56 (*p* = 0.01)	*r* = 0.83 (*p* < 0.001)	*r* = 0.27 (n.s.)	*r* = 0.03 (n.s.)	*r* = 0.37 (n.s.)	*r* = 0.05 (n.s.)
significance of change	*p* < 0.05	*p* < 0.05	*p* < 0.001	*p* = 0.001	*p* = 0.001	*p* < 0.001

This table shows the correlations and changes of different parameters at baseline and after adrenalectomy in patients with aldosterone-producing adenomas. Abbreviations: ARR—aldosterone to renin ratio, AUC—area under the curve, BP—blood pressure, SUSPPUP—serum sodium to urinary sodium divided by (serum potassium)^2^ to urinary potassium ratio.

**Table 6 diagnostics-10-00545-t006:** Influence of antihypertensive medication on parameters in different patient groups.

Medication		K_S_ w [mmol/L]	K_S_ w/o [mmol/L]	ARR w	ARR w/o	SUSPPUP w [L/mmol]	SUSPPUP w/o [L/mmol]	ARR×SUSPPUP w [L/mmol]	ARR×SUSPPUP w/o [L/mmol]
**BBL**	in PA	3.5 ± 0.5	3.6 ± 0.5	65.2 (40.0–129.6)	76.1 (41.6–136.6)	7.30 (4.2–12.1)	8.2 (4.0–13.6)	495 (228–1090)	498 (257–1499)
	in EH	4.3 (4.1–4.6)	4.3 ± 0.3	8.5 (2.4–21.7)	10.0 (1.8–23.9)	3.3 (2.4–4.7)	2.9 (2.3–4.3)	30 (8–104)	28 (5–73)
**ACEI**	in PA	3.5 ± 0.6	3.6 ± 0.5	62.8 (34.4–78.2)	81.8 (42.3–164.6)	8.9 (3.9–15.5)	7.3 (4.3–11.2)	440 (235–1112)	503 (228–1717)
	in EH	4.2 (4.0–4.6)	4.3 ± 0.4	**1.3** **(0.7–9.0)**	**12.2** **(6.2–26.3)**	3.2 (2.3–4.0)	3.2 (2.3–4.5)	**5** **(1–26)**	**46** **(18–226)**
**ARB**	in PA	3.5 ± 0.6	3.6 (3.3–3.9)	74.0 (44.1–148.2)	71.1 (40.1–129.4)	7.3 (5.4–13.3)	7.4 (4.0–12.4)	495 (312–1637)	498 (196–1084)
	in EH	4.3 ± 0.6	4.3 ± 0.3	**22.1** **(7.3–61.6)**	**8.5** **(1.8–20.1)**	3.5 (2.3–4.4)	2.9 (2.1–3.4)	52 (13–167)	21.4 (6–73)
**diuretics**	in PA	3.5 ± 0.6	3.6 (3.5–3.9)	67.8 (36.6–161.2)	74.0 (42.1–113.2)	8.2 (4.8–12.7)	7.3 (3.9–12.7)	510 (228–1951)	485 (266–1066)
	in EH	**4.1 ± 0.5**	**4.4 ± 0.3**	5.7 (1.5–29.2)	9.6 (2.2–22.1)	3.1 (2.4–3.8)	3.2 (2.3–4.5)	20 (5–100)	30 (9–194)

Results are presented as means ± standard deviation if normally distributed or medians (interquartile range). Figures in bold indicate statistical significance with a *p*-value of less than 0.05. Abbreviations: ACEI—angiotensin-converting enzyme inhibitors, ARB—angiotensin II type 1 receptor blockers, ARR—aldosterone to renin ratio, BBL—betablockers, K_S_—serum potassium, SUSPPUP—serum sodium to urinary sodium divided by (serum potassium)^2^ to urinary potassium ratio, w—with, w/o—without.

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
