# Peer review of "Aldosterone-Mediated Sodium Retention Is Reflected by the Serum Sodium to Urinary Sodium to (Serum Potassium)2 to Urinary Potassium (SUSPPUP) Index"

_diagnostics, 2020, doi:10.3390/diagnostics10080545_

Round 1
Reviewer 1 Report
Kanaan E et al reported that SUSPPUP ratio that to can be obtained by simple and easy testing helps to improve the diagnosis of hyperaldosteronism. Their findings might be so useful for generalists.
- They showed only a combination of ARR and SUSPPUP ratio.
How is ARR × potassium or SUSPPUP ratio × potassium?
- TTKG and FEK are also well known for an index of the potassium excretion ability. Is SUSPPUP ratio useful in comparison with these indexes?
- Do SUSPPUP ratio or SUSPPUP × ARR correlate with PA subtype (aldosterone-producing adenoma or idiopathic hyperaldosteronism)?
- Figure 3 showed that SUSPPUP ratio was significantly decreased on anti-MC treatment than basal condition. Is SUSPPUP ratio useful for an index of the effect of anti-MC? Or, does this ratio correlate with the onset of complications?
- Many patients with hypokalemia due to hyperaldosteronism are tend to receive the replacement of potassium. Does the replacement therapy with the potassium preparation have an influence on this ratio?
Author Response
How is ARR x potassium or SUSPPUP ratio x potassium
The combination of ARR ´ potassium is indeed a very good idea. Since its performance seems to be better than ARR only, this became matter of a larger study which includes more centers. The study, however, is not yet finished. For your information, it was not possible to built a likewise large base for investigating the SUPPUP index because urinary samples are less frequently analyzed in such patients and much more dependent on diagnostic SOPs. Potassium, however, is available along with the ARR. We can give our results on ARR ´ K for this study but would like to take caution that more power be achieved with a different study design. If possible, we would like to forgo this analysis.
The combination of SUSPPUP x potassium loses power (Willenberg/Kolentini/Quinkler et al. Eur J Clin Invest 2009) and was by far not as useful as the SUSPPUP x ARR calculations.
TTKG and FEK are also well known for an index of potassium excretion ability. Is SUSPPUP ratio useful in comparison with these indixes?
For measurements of TKK or FEK, other parameters are necessary which were – for what reason ever – not available for every case. Osmolality is not part of the protocol, while creatinine actually is but was not determined in each case since GFR calculation are nowadays done using other formulae. Therefore, TKK can not be determined with our data set, FEK only in a subset of patients. FEK was measured in dogs with babesiosis (Texas fever) and found to be similarly altered as the SUSPPUP ratio and the authors thought of mineralocorticoid excess. However, as cells die potassium is excreted through the kidney. Therefore, we would like to not cite this work (Zygner W et al. Changes in the SUSPPUP ratio and fractional excretion of strong monovalent electrolytes in hospitalized dogs with canine babesiosis. Pol J Vet Sci 2012, 15, 791-792).
Do SUSPPUP ratio or SUSPPUP x ARR correlate with PA subtype?
Results from calculations of SUSPPUP or SUSPPUP x ARR rather seem to correlate more with severity than with PA subtype. However, the data is insufficient to support this claim. We have added the patient number with unilateral disease in the Results section and indicated where the number of correctly identified patients was higher in the group with unilateral PA (which was statistically only the case for potassium). This may be because ARR and SUSPPUP ´ ARR are very good in identifying individuals affected by PA. A much larger patient number would needed to prove that these indices differentiate between PA subtype (overlap problem which exists for good sensitivity). We have added these results and thoughts.
»Patients with unilateral PA were significantly more likely to experience hypokalemia than PA patients with bilateral disease (81.6 % vs. 58.1 %, p<0.05).«
»ARR x SUSPPUP was elevated in 94.7 % of individuals with unilateral and in 90.3 % with bilateral PA (p=n.s.), the ARR detected 92.1 % unilateral and 80.6 bilateral PA patients (p=n.s.), and the SUSPPUP index 71.1 % patients with unilateral vs. 67.7 % with bilateral disease (p=n.s.).«
Is SUSPPUP ratio or SUSPPUP x ARR useful for an index of the effect of anti-MC? Or does this ratio correlate with the onset of complications?
An impaiment in kidney function may well distort our results. As such, we interprete our data more in the light as the SUSPPUP being an index of MC function (or treatment) rather than as an indicator of complications. However, the latter aspect can not really be shown by us. Therefore, we added the cautionary note on kidney function to the Discussion section.
»Whether SUSPPUP ratio or potassium maintain their informative power in patients with PA and kidney injury remains open but seems likely in patients with impairment in renal electrolyte handling.«
Does the replacement therapy with potassium preparation have an influence on this ratio?
The treatment with potassium will likely impact on renal potassium excretion. Therefore, patients with potassium treatment during urinary sampling have not been included. This aspect was added to the Methods section:
»Patients had been asked to stop supplementation with potassium during urinary sampling.«
From experience (not from this study) it is to note that urinary potassium excretion rises during supplementation with potassium and may create false positive results in individuals w/o mineralocorticoid excess.
Potassium supplementation during the screening with the ARR is one aspect why we started to look into the performance of the combination of ARR x potassium in comparison to the ARR in a different study (results are not ready yet).
Reviewer 2 Report
In the present study, the authors evaluated the PA diagnostic ability of SUSPPUP, an indicator of sodium retention and potassium excretion.
First, they found that the diagnostic ability of hypokalemia is best when the threshold is less than 3.9mmol/l. Secondly, SUSPPUP alone has inferior diagnostic performance to hypokalemia, defined as less than 3.9mmol/l, while the diagnostic ability of the product of SUSPPUP and ARR exceeds that of hypokalemia alone. Furthermore, they showed that the product of SUSPPUP and ARR was less sensitive to antihypertensive drugs, especially diuretics and beta-blockers, and this indicator changes in response to PA-specific treatments. In conclusion, they described this indicator can improve diagnostic ability of PA.
The topic is of interest. However, there are some issues to be considered.
1.Patients and Methods
Is this study really a prospective study when you incorporated patients from existing registries? Name the patient enrollment period.
2.Patients and Methods
Related to what I mentioned above, Was the EH group really “essential” ? The resistry used in this study appears to be essentially a registry of patients with adrenal disease, or endocrine disorders such as pituitary or gonadal. Is it appropriate as a patient population for a study to evaluate the diagnostic ability of PA? Were patients with adrenal incidentaloma, other pituitary or gonadal diseases included? Do those diseases really have no effect on blood pressure? Did you perform dexamethasone surpression test on all cases with adrenal tumor? Please mention about these things and include them as “Limitation” for this study.
3.Patients and Methods
This population had a small number of EH patients and is disconnected from real world practice. Please mention about these things and include them as “Limitation” for this study.
4.Table3
You described that urinary sodium in PA patients was significantly lower than that in other patients(Line 80), but Table 3 does not list it, and instead, serum sodium. Please list urinary sodium and potassium in Table 3.
5.Figure 1
Can you consider the PA diagnostic ability of ARR combined with serum potassium ? Is diagnostic ability of the product of ARR and SUSPPUP superior to that of ARR combined with serum potassium ? I think it is important to consider the significance of SUSPPUP.
6.Figure3
Were anti-MC therapy administered only for PA patients or including EH patients? You should be clearly stated about that. The difference of SUSPPUP between anti-MC therapy cases and surgical cases could be a matter of anti-mineralcorticoid drugs dosage or PA subtype. You should describe the percentage of subtypes of PA patients treated with anti-MC therapy. You also add the dose of anti-mineralcorticoid drugs and the plasma renin activity value after administration.
7.Conclusion
In this study, you didn’t describe anywhere that SUSPPUP show better diagnostic ability in the patients with hypokalemia. You describe only that the diagnostic ability of hypokalemia, defined as less than 3.9mmol/l, is relatively high. You should remove following statement, “especially if hypokalemia is defined as a serum concentration <3.9 mmol/l.”.
Author Response
1.-3. Patient and Methods
The study design of both the Conn registry and the SHIP-PAGE study allows for the inclusion of patients with PA or who came with a suspicion for PA or for exclusion of PA. A retrospective arm and a prospective arm are both implemented; the idea of this very investigation was that of a prospective study, which started at our institution in Dec 2007 (ending 2013). The dates have been added to the methods section:
»Blood and 24-hour urine samples were collected from patients who gave their informed consent between december 2007 and december 2013.«
However, the design explains the relative low number of patients with EH (your comment), a fact which comes with limitations – unfortunately. This was stated in the Discussion section:
»As a limitation, the dataset is manageable if it comes to the number of patients with essential hypertension which is rather small in comparision to their share in the general population or primary care settings which is due to the study design.«
It is possible to also study patients with other endocrinopathies under the umbrella of the the SHIP-PAGE study. However, pituitary, gonadal, or apperent Cushing patients have not been included or used as reference individuals. Dexamethasone suppression tests have been performed in almost all patients (except if patients declined); patients with adrenal Cushings or relevant co-secretion of cortisol (for example low basal corticotropin) have been excluded. This information was added to the manuscript:
»Cushing’s syndrome or relevant co-secretion of cortisol was ruled out if a dexamethasone test rendered a normal result of cortisol <50 nmol/l (1,8 µg/dL).«
4. Table 3
The statistical analysis for urinary sodium »NaU« was added.
5. Figure 1
The combination of ARR x potassium is a very good idea. Since its performance seems to be better than ARR only, this became matter of a larger study which includes more centers. The study, however, is not yet finished. For your information, it was not possible to built a likewise larg base for investigating the SUPPUP index because urinary samples are less frequently analyzed in such patients and much more dependent on diagnostic SOPs. Potassium, however, is available along with the ARR. We can give our results on ARR x K for this study but would like to take caution that more power be achieved with a different study design. If possible, we would like to forgo this analysis.
6. Figure 3
The point raised by the reviewer is important. Since therapy with spironolactone, eplerenone, triamterene or amiloride was limited to single cases with EH we skipped that analysis and patients were asked to stop this medication 4 to 6 weeks prior to investigation anyway.
Therefore, only PA patients were considered in this very analysis which is stated now in the legend:
»Panels show the change in the SUSPPUP ratio (a), serum potassium (b), the ARR (c) and ARR x SUSPPUP (d) from baseline, over anti-mineralocorticoid (MC) treatment to a visit between 1 and 3 months after adrenalectomy in patients with PA.«
There is, however, a problem with anti-MC dosage. This is because because patients receive sprionolactone, males rarely more than 25 mg, and this regimen is supported with eplerenone, triamterene, and/or amilorid. In the majority of cases, amiloride or triamterene can not be given as single compounds in Germany and then are available in combinations with hydrochlorothiazide which is not a potassium-sparing antimineralocorticoid diuretic but – nevertheless – helps to lift (and treat against) the mineralocorticoid pressure. In our experience, there is no good method of calculating an equivalence dose because WHO's daily defined dosages were meant for patients with (and data is only known from patients with) essential hypertension. Therefore, dosage of anti-MC medication is done step-wise and with combinations until renin becomes measurable or increases and hypokalemia is absent. We added renin values for your and reader's information in the legend of Figure 3:
»Plasma renin concentrations were 5.7±5.7 pg/ml on anti-MC treatment (please, see also the Patients and Methods section).«
and explained our procedure in the Methods section for better traceability:
»…as well after an antimineralocorticoid treatment with spironolactone, eplerenone, triamterene, or amilorid – the latter two also in combination with hydrochlorothiazide. Since WHO's daily defined doses are not suitable to calculate equivalent doses in patients with PA, the procedure is to start with 25 mg of spironolactone, followed by an increase to 50 mg in the absence of side effects or a combination with amiloride mono whenever possible or in combination with a thiazide if a switch to eplerenone is not possible. In many such cases, family doctors then frequently change to triamterene/thiazide – starting at 25/12.5 mg which we increased to 50/25 mg if renin remains low or hypokalemia persists.«
7. Conclusion
The phrase was canceled.
Round 2
Reviewer 2 Report
Authors responded to comments appropriately.
There is no additional comments.